# Conversion of CO_2_ into Chloropropene Carbonate Catalyzed by Iron (II) Phthalocyanine Hypercrosslinked Porous Organic Polymer

**DOI:** 10.3390/molecules25204598

**Published:** 2020-10-09

**Authors:** Eva M. Maya, Antonio Valverde-González, Marta Iglesias

**Affiliations:** Instituto de Ciencia de Materiales de Madrid (ICMM), Consejo Superior de Investigaciones Científicas (CSIC), c/Sor Juana Inés de la Cruz 3, Cantoblanco, 28049 Madrid, Spain; a.valverde.gonzalez@csic.es (A.V.-G.); marta.iglesias@icmm.csic.es (M.I.)

**Keywords:** CO_2_ conversion, cyclic carbonate, knitted porous polymer, iron (II) phthalocyaninates, recyclability

## Abstract

Commercial iron (II) phthalocyanine (FePc) was knitted with biphenyl using a Friedel–Crafts reaction to yield a micro-meso porous organic polymer (FePc-POP) with a specific surface area of 427 m^2^/g and 5.42% of iron loading. This strategy allowed for the direct synthesis of a heterogeneous catalyst from an iron containing monomer. The catalytic system, formed by the knitted polymer containing FePc and DMAP (4-dimethylamino pyridine) as base, results in an efficient heterogeneous catalyst in the cycloaddition of CO_2_ to epichlorohydrin to selectively obtain the corresponding cyclic carbonate. Thus, a TON (mmol substrate converted/mmol catalysts used) value of 2700 was reached in 3 h under mild reaction conditions (solvent free, 90 °C, 3 bar of CO_2_). The catalyst does not exhibit leaching during the reactions, which was attributed to the excellent stability of the metal in the macrocycle.

## 1. Introduction

The cycloaddition of CO_2_ to epoxides is a well-known reaction that has aroused great interest, especially in recent years, because it is a safe and fully atom-economical way to produce cyclic carbonates or polycarbonate, taking advantage of a harmful gas such as CO_2_.

The selectivity between cyclic carbonate and polycarbonate can be adjusted through the reaction temperature and CO_2_ pressure. Thus, higher reaction temperatures will promote the formation of the cyclic carbonate which is the thermodynamically favored, whereas lower reaction temperatures use to yield polycarbonate. Furthermore, higher CO_2_ pressure improves the contact between the reagents enhancing the rate of the CO_2_-insertion step causing the formation of the polycarbonate product [1].

Thus, a great variety of homogeneous and heterogeneous catalysts have been explored to carried out this reaction to selectively obtain cyclic carbonates [2,3], which are extensively used as solvents to remove oils, as intermediates in organic synthesis or as monomers of plastics’ engineering [4].

However, to endorse a sustainable application, the use of heterogeneous metal based catalysts is preferred. Generally, these types of catalysts have good catalytic activity, require low catalysts loadings, and benefit from easy separation and recyclability due to their heterogeneous nature. Among the heterogeneous metal based catalysts that afford this transformation, those containing chromium, nickel, zinc, copper, cobalt and aluminum are the most studied [5,6,7]. However, iron could be considered as the optimum choice from a sustainable approach because it is cheap, earth abundant, and has relatively low toxicity [8]. Thus, in the last few years, some heterogeneous iron catalysts, that effectively promote this reaction, have been reported, e.g., iron doped mesoporous zeolites [9,10], a porous covalent triazine-based polymer containing iron oxide nanoparticles [11] and some iron-based poly(azomethine)s [12,13,14]. All reactions carried out with iron-containing heterogeneous catalysts produced cyclic carbonates in high yields, using CO_2_ pressures between 3 and 20 bar and reaction temperatures in the range of 50–120 °C [9,10,11,12,13,14]. The use of metal containing porous organic polymers (POPs) for this application offer some advantages over other materials because, in addition to their tailored porosity, they have an affinity for CO_2_ and a predominant covalent character which offers high thermal and chemical stability [15]. However, in most of the metal containing POPs reported, the metal is incorporated by post-functionalization of the previously formed network [16,17,18], which can favor the possibility of metal leaching, questioning the heterogeneity of the system. Some metal based POPs have been prepared from monomers that already contain metal with good results in this transformation and excellent recyclability, such as POPs containing metalated porphyrins [19,20,21,22,23].

However, no example has been reported to accomplish this conversion using a metal phthalocyanine based POP. This macrocycle offers unique optical and electronic properties besides of an excellent thermal and chemical stability [24]. Moreover, the metal loss in phthalocyanine rings only occur under certain conditions. For example, the demetalation of MgPc can be carried out in a quartz cuvette in a degassed dimethylformamide solution using an excess of 1,4-dihydroxyanthraquinone or 1,2,4-trihydroxyanthraquinone in the dark [25]. A partial elimination of nickel in a NiPc was achieved by halogenation using NaBr followed by addition of oxone solution [26], whereas ZnPc was also partially demetaled under a mixture of pyridine and HCl at 120 °C for several hours [27]. Recently, it was observed that iron was lost in a FePc during an oxygen reaction reduction process [28]. However, when iron containing POPs were used in ORR reactions by us [29], no iron loss was observed, which proves the high metal stability of these materials.

Thus, in this work, we report for the first time the catalytic activity of an iron (II)-phthalocyanine porous organic polymer in the cycloaddition of CO_2_ to epoxides, under mild conditions.

## 2. Results and Discussion

### 2.1. Synthesis and Characterization of Iron (II) Phthalocyanine Porous Organic Polymer (FePc-POP)

The heterogeneous catalyst based on iron (II) phthalocyanine (FePc-POP) was prepared following the knitting solvent strategy (Scheme 1) which involves a Friedel–Crafts reaction where either an external cross-linker or the solvent acts as a linker between the phenyl groups of the monomers (FePc and biphenyl in this case) [29]. This procedure known as the knitting solvent strategy consists of a reaction between aromatic monomers and a cross-linker that joins the molecules and it was first reported in 2011 by Bien Tan [30]. Since then, there is only one example of a phthalocyanine-containing porous organic polymer, an aluminum (III) phthalocyanine which, after its polymerization, was treated with an excess of K[Co(CO)_4_] to obtain the corresponding active catalyst in the carbonylation of epoxides with CO [31].

Thus, iron (II) phthalocyanine and biphenyl were reacted in dichloroethane using formaldehyde dimethyl acetal as cross-linker and FeCl_3_ as catalyst. The reaction was kept at 80 °C for 48 h to yield FePc-POP [29]. The catalyst was fully characterized using the following techniques: TXRF to determine the amount of iron, elemental analysis to obtain the C, H and N content; FT-IR (Appendix A), Raman (D), and UV-Vis-NIR (Appendix A) spectroscopies to confirm the presence of the FePc units in the network; solid state ^13^C-NMR (Appendix A) to ratify the connection between the aromatic rings through -CH_2_- bonds; TGA to show the thermal stability (Appendix A); N_2_ adsorption-desorption isotherms to obtain the porosity parameters (Appendix A); SEM microscopy to observe the morphology (Appendix A) and EDX to confirm the homogeneous dispersion of iron in the sample (Appendix A). The relevant characterization data are recorded in Table 1.

### 2.2. Catalytic Activity

As mentioned in the introduction, the new catalyst was employed in the cycloaddition of CO_2_ to epoxides, an elegant strategy to convert this gas in useful compounds such as cyclic carbonates. The optimal reaction conditions were established using epichlorohydrin (ECH) as substrate and dimethylamino pyridine DMAP as base. Epiclorohydrin was selected as model substrate because is one of the most interesting epoxides that can be produced from glycerin obtained from vegetable oil [32] and DMAP as base due its proved efficiency compared with other bases [12,33,34,35]. The excellent catalytic activity in presence of this base was firstly reported by Paddok et al. [36] who attributed it to the coordination of DMAP to the metal generating more electron-rich centers in the metal-catalyst.

The first reaction (entry 1, Table 2) was done at 3 bar of CO_2_ and 70 °C in absence of solvent with an ECH/DMAP/metal ratio of 1000:10:1. Under these reaction conditions, a 59% of ECH conversion was achieved with the selective formation of the corresponding cyclic carbonate, chloropropene carbonate. Thus, in order to improve the conversion, the temperature was increased to 90 °C, and after 3 h, the ECH conversion reached 94% (entry 2). These results highlighted the need to heat up to 90 °C to obtain reasonable epoxy conversions. To increase the turn over frequency (TON) of the reaction (mmol substrate converted/mmol catalysts used), the amount of catalyst was reduced. Thus, using a relationship ECH: Fe of 2000 (entry 3) the TON raised to 1860 and using a ratio ECH: Fe of 3000 (entry 4) the TON reached a value of 2700 with ECH conversions of 93 and 90%, respectively. These results indicated that small amount of catalyst system is sufficient to carry out this reaction. The molar ratio substrate: DMAP: catalysts (based on iron) of 3000:10:1 was established as the most optimum for this reaction. It is important to point out that in all cases only chloropropene carbonate (Selectivity > 99%) was obtained, which highlighted the high selectivity of this catalytic system.

To evaluate the effect of the base and FePc-POP in the reaction, two control experiments were done. Firstly, the reaction was carried out in the absence of FePc-POP (entry 5) using the same epoxy: DMAP ratio and conditions of entry 4. As can be observed, a 30% of tAhe ECH conversion can be attributed to the DMAP, as was previously observed for the same reaction in similar conditions [12]. Using only FePc-POP as catalyst in the same conditions (entry 6), a small amount of ECH was converted into cyclic carbonate. These results indicated an important cooperative effect between the FePc-POP catalysts and the base, since together yielded the highest conversion. This cooperative effect was previously reported for other metal heterogeneous catalyst-DMAP systems [12,13,14,36]. Additionally, the reaction was carried out using FePc as homogeneous catalyst (entry 7). The ECH conversion was low, 33%, similar to the conversion attributed to the base, which indicated very low catalytic activity of the FePc monomer. This result could be attributed to the great aggregation tendency of the phthalocyanine rings which difficult the accessibility to the iron to promote the reaction. This result also confirms the advantage to polymerize this monomer, migrating from a very bad to an excellent catalyst.

When styrene oxide was used as substrate lower conversions were achieved (entry 8), even with longer reaction times (TON 870). The lower reactivity of this substrate was previously attributed to electronic factors rather than to steric ones [37]. The use of an internal epoxide, such as cyclohexene oxide (CHO) (entry 9), did not lead to good results due to the difficulty in opening the epoxide caused by the steric hindrance between the two rings.

The catalytic activity of FePc-POP was compared with those reported for some heterogeneous porphyrin- [19,20,21,22,23,38,39,40,41] or phthalocyanine-based [42,43] catalysts in the cycloaddition of CO_2_ to ECH (Table 3). The Mg porphyrin based catalysts (entries 1 and 2) work with very high substrate/metal ratios which result in very high TON values but higher CO_2_ pressures and temperatures are needed to achieve total conversion. The Al-porphyrin based material Al-iPOP (entry 4) requires a lower substrate/metal ratio than in our system, as well as lower temperature but higher CO_2_ pressure. Co-POP TPP/TBAB (entry 5) works under ambient conditions, and some others porphyrin-based catalysts operate at lower pressures (entries 7–9), needing a longer reaction time and higher catalyst amount than ours, yielding lower TON values. Comparing with the phthalocyanine- based catalysts (entries 10 and 11) it can be seen that our catalytic system is used in less amount and the reaction occurs at shorted reaction times.

In summary, compared to the current literature, our phthalocyanine-based/DMAP catalyst is highly competitive in terms of both reaction times and catalyst loading.

### 2.3. Recyclability

The recyclability of FePc-POP was also studied using epichlorohydrin (ECH) as a substrate at a ratio 1000:1, to handle a greater amount of catalyst and facilitate its recovery. After 2 h, the reaction was stopped, and the liquid was removed. The catalyst was centrifugated with ethanol three times and then it was dried overnight at 90 °C. The liquid was analyzed by gas chromatography observing an ECH conversion of 85.4% (Figure 1 left, run 1). The catalyst was weighed and a new reaction was done using the corresponding amount of fresh ECH and DMAP. After 2 h the ECH conversion was 80.5% (run 2). The procedure was applied three more times (runs 3–5), observing a slight decrease in the ECH conversion. The IR spectrum recorded after the fifth run (Figure 1 right) showed the characteristics FePc bands between 1330 and 730 cm^−1^ and those attributed to the biphenyl groups. However, the spectrum showed also a broad adsorption at 1795 cm^−1^ that could be ascribed to some C=O groups from the partial oxidation of methylene linkages during the reaction. The presence of these groups could be causing the decrease in the ECH conversion.

To confirm the heterogeneous nature of FePC-POP, a hot filtration experiment was done (Figure 2). Thus, in a new experiment using ECH as substrate, DMAP and the catalyst in the ratio 3000:10:1, the reaction was stopped after 1 h obtaining a 45% of ECH conversion. The catalyst was removed from the reaction which was running for 2 more hours in the absence of catalyst, reaching 57% of ECH conversion. This increase was attributed to the DMAP than can promote by itself the reaction, as it was previously observed in this work and also reported [12,14]. We have analyzed the iron content in the filtrate, after the catalyst removal, by ICP-MS and also in the filtrate of the runs of the recycling experiments. The very low iron content of less than 3 ppm suggests that the scarce leaching of the catalyst affords inactive iron derivatives for this reaction. In conclusion, we can guarantee that the heterogeneous nature of the catalyst is maintained even after the recycles. These results are in agreement with the difficulty of metal phthalocyanines to lose the metal.

## 3. Materials and Methods

### 3.1. Materials

Biphenyl was purchased by Fluka with a purity > 98%; Iron (II) phthalocyanine (FePc) was supplied from TCI with a purity > 98%; Anhydrous FeCl_3_ was obtained from Alfa Aesar with 98% of purity; and CH_2_Cl_2_ (extra dry) was supplied by Across Organics with purities > 99.8% respectively. The other reagents and solvents were supplied by Sigma-Aldrich San Luis, MO, USA) with analytical grade. FePc-POP was prepared following the procedure reported [29].

### 3.2. Characterization

Elemental Analysis (C, H, N) were made using a Carlo Erba EA1108 elemental analyzer (CE Instruments Ltd., Wigan, UK). The iron content was determined with a abenchtop S2 PicoFox TXRF spectrometer from Bruker Nano (Bruker Corporation, Billerica, MA, USA), equipped with a multilayer monochromator with 80% of reflectivity at 17.5 keV (Mo Kα), a Mo X-ray source working at 50 kV and 600 μA, and a XFlash SDD detector with an effective area of 30 mm^2^. ^13^C CP-MAS spectrum was recorded in a Bruker AV-400-WB (Bruker Corporation, Billerica, MA, USA) with a spectral width of 35 kHz and the chemical shift refers to 29.5 ppm (the CH_2_ signal of adamantine) as a secondary reference, and to TMS as the primary reference. Fourier Transform Infrared Spectra (FTIR) were recorded on Bruker iFS 66VS Spectrometer (Bruker Corporation, Billerica, MA, USA). UV-Vis Diffuse reflectance spectra were carried out on a Shimadzu UV-2401 PC UV-Visible Spectrometer (Shimadzu Corporation, Kioto, Japan). Nitrogen adsorption isotherms was measured using a Micromeritics ASAP 2020 M surface at 77 K (Micromeritics Instrument Corporation, Norcross, GA, USA). The samples were degassed for 12 h at 100 °C before taking the measurement. Specific surface area was determined by the BET technique and the pore size distribution by NLDFT methods. Scanning electron microscopy (SEM) image was acquired from a Hitachi SU-8000 microscope (Hitachi Ltd. Tokio, Japan) operating at 0.5 kV. EDX-analysis was complete using a metalized by gold sample in a FEI microscope NOVA NanoSEM 230 model provided with an EDAX-Ametek detector (FEI Company, Hillsboro, OR, USA).

### 3.3. Catalytic Activity

General procedure: 6.5 mmol of epoxide, 0.65 mmol of biphenyl (internal standard), dimethylaminopyridine (DMAP) and FePc-POP in the ratios (500–3000:10:1, depending on the procedure used) were placed in a Picoclave Buchi glass reactor of 10 mL. Then, the reactor was charged with CO_2_ at the pressure indicated in each experiment. The reaction mixture was stirred at the temperature and time indicated in Table 2.

All reactions were monitored by gas chromatography-mass spectrometry on a Konik HRGC 4000B GC-MS chromatograph (Konik Instrument Int, Miami, FL, USA) with a cross-linked (95%)-dimethyl-(5%)-diphenylpolysiloxane (Teknokroma TRB-5MS) column of 30 m of length; helium as carrier gas, 20 psi; injector temperature: 230 °C; detector temperature: 250 °C. Oven program for the reaction: 70 °C (5 min), 9 °C min^−1^ to 220 °C (10 min), retention times: Biphenyl 17.2 min; ECH: 3.09 min ECH-Cyclic Carbonate: 13.7 min; SO: 9.1 min, SO-Cyclic Carbonate: 20.5 min; CHO: 5.5 min, CHO-Cyclic Carbonate: 11.8 min.

Recycling experiments: 1.5 mL (0.019 mol) of ECH, 200 mg (0.00129 mol) of biphenyl, 15.84 mg of FePc-POP (0.019 × 10^−3^ mol of Fe) and 20.7 mg (0.019 × 10^−2^ mmol) of DMAP were added to the Picoclave Buchi glass reactor. Then the reactor was charged with 3 bar of CO_2_ and stirred at 90 °C for 2 h. Then the liquid was removed with a pipette and the catalyst was centrifuged with 5 mL of ethanol (three times). The ethanol was removed, and the catalyst was dried at 90 °C overnight. After weighing the amount of catalyst recovered, biphenyl, fresh ECH, and DMAP were added to the reactor, maintaining the starting ECH:DMAP:Fe ratio, and it was charged again with 3 bar of CO_2_. After using the same protocol to recover the catalyst, the procedure was repeated 3 more times. Each reaction was monitored by GC-MS using the above oven program.

## 4. Conclusions

A system formed by a knitted polymer containing FePc and biphenyl (FePc-POP) and DMAP is an excellent heterogeneous catalyst in the conversion of CO_2_ to selectively obtain chloropropene carbonate from epichlorohydrin. A TON value of 2700 was reached in 3 h, in a solvent free reaction under mild conditions, namely 90 °C and 3 bar of CO_2_. The FePc rings provide great stability to the metal in the catalyst avoiding its leaching to the medium. It is important to highlight that the new heterogeneous catalyst system (FePc-POP/DMAP) is the only one responsible for the catalytic activity since the control experiments carried out using the FePc molecule as catalyst with or without DMAP did not produce any cyclic carbonate.

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
