# Peer review of "Conversion of CO2 into Chloropropene Carbonate Catalyzed by Iron (II) Phthalocyanine Hypercrosslinked Porous Organic Polymer"

_molecules, 2020, doi:10.3390/molecules25204598_

Round 1

Reviewer 1 Report

In this manuscript, authors describe the preparation of a FePc-POP catalyst and its application to the thermal and photocatalytic conversion of CO2 into cyclic carbonates. The synthetic procedure consists of a first step in which monomers of FePc are transformed into a porous organic polymer (FePc-POP) through a Friedel-Crafts reaction. This catalyst is directly applied in the thermal cycloaddition of CO2 to epoxides. In order to increase the sustainability of the process, FePc-POP is also supported on TiO2 to obtain a hybrid photocatalyst capable of driving the reaction under illumination with a LED lamp.  The as-prepared catalysts have been characterized using several techniques including Raman spectroscopy, elemental analysis, TGA, FT-IR and UV-vis absorption spectroscopy. Authors have demonstrated that, under optimal conditions, the FePc-POP catalyst is able to conduct the thermal conversion of CO2 into cyclic carbonates under mild conditions. Besides, the catalyst can be recycled upon consecutive uses. The hybrid FePc-POP/TiO2 photocatalyst has also shown catalytic activity towards the cycloaddition of CO2 to epoxides.

The use of organic dyes as sensitizers to enhance the activity of wide bandgap semiconductors like TiO2 has been widely studied since many years ago, especially in the field of photovoltaics and photocatalysis. Traditionally, the main issue of this approach is the low stability of organic molecules under reaction conditions. In this work, authors have used this strategy to synthesize and hybrid photocatalyst for the conversion of CO2 into cyclic carbonates under light radiation. The thermal conversion of CO2 using the as-prepared FePc-POP also renders good catalytic performance at relatively low temperature and reaction time. In spite that the photochemical mechanism should be discussed in more detail, overall, the quality and novelty of the work are adequate and acomplish the standards required for this journal.

For these reasons, I would recommend for its publication in Molecules. However, before its publication I would like to make some suggestions to further improve the quality of this work.

Major revisions:

  • Authors should determine the amount of Fe both in the FePc-POP catalyst and in the liquid after thermal reaction. Hot filtration test reveals a residual catalytic activity that authors attribute to DMAP itself, however, this can also be due to the presence of Fe in the reaction mixture. This leaching could also explain the slight deactivation upon reuses.  
  • Overall, the characterization of FePc-POP/TiO2 hybrid photocatalyst is poor and it should be improved. UV-vis spectra of FePc-POP, TiO2 and hybrid FePc-POP/TiO2 should be included as a figure in the manuscript. XRD analysis and electron microscopy images with EDX mapping would help to have a better knowledge of the hybrid structure. In fact, authors have included SEM in “Materials and methods” section, but no SEM image is present in the manuscript.
  • Authors have demonstrated the photocatalytic activity of the FePc-POP/TiO2 hybrid catalyst towards the synthesis of cyclic carbonates using CO2. According to blank experiments, hybrid catalyst features a much higher activity than the individual elements, thus indicating a synergy between them. However, they do not provide a reaction mechanism that clarifies which is the role of each of the components of the photocatalyst. Why is the photocatalytic activity of FePc-POP enhanced when supported on TiO2? What is the role of light in the process? Authors should explain in more detail.
  • Have the authors measured the temperature of the reactor under illumination? Authors should perform a blank experiment under this temperature and dark conditions to exclude a possible thermal pathway induced by light heating.
  • Is it possible to recycle the FePc-POP/TiO2 hybrid photocatalyst? Authors have not included any recycling test in the study to evaluate the stability of this material. This is from pivotal importance, as stability issues are common in this type of hybrid organic-inorganic structures under reaction conditions.
  • The amount of FePc-POP in the hybrid photocatalyst has not been optimized. According to the authors, FePc-POP loading on TiO2 is 10%, have they tried with higher or lower loadings? Have they quantified the real amount of Fe in the samples?
  • Why authors use DMAP as base in the reaction instead of TBAB, as previously reported in the literature? What is the advantage of using DMAP?

Minor revisions:

Authors should revise the manuscript and correct some typos and grammatical mistakes:

  • Line 66: “a hybrid catalyst”
  • Line 68: “using a 20 W white LED”
  • Line 69: please define MOF if it is the first time you use this abbreviation.
  • Line 75: “Thus, in this work we report for the first time”
  • Line 124, reaction scheme: The abbreviation of cyclohexene oxide is incomplete. Please exchange “CH” by “CHO”.
  • Line 234: “was stopped”
  • Line 247: Please revise purity of FeCl3, there is a possible typo.
  • Line 253: “were mixed”
  • Line 276: “indicated”
  • Line 277: Please provide lamp specifications.

Author Response

Major revisions:

  • Authors should determine the amount of Fe both in the FePc-POP catalyst and in the liquid after thermal reaction. Hot filtration test reveals a residual catalytic activity that authors attribute to DMAP itself, however, this can also be due to the presence of Fe in the reaction mixture. This leaching could also explain the slight deactivation upon reuses.

The amount of iron in the catalyst before and after the reaction is the same, since phthalocyanine macrocycles are very stable and cannot be demetaled in the conditions of our reaction. The demetalation reactions of metal phthalocyanines only occur under certain conditions. For example the demetalation of MgPc can be carried out a degassed DMF solution with an excess of 1,4-dihydroxyanthraquinone [Int. J. Phtotoenergy, 2000, 2(1), 9-15]; partial demetalation of NiPc can be achieved by halogenation of using NaBr followed by addition of oxone solution [Fuel, 2015, 161, 43-48] or ZnPc can be also partially demetalated under a mixture of pyridine and HCl at 120ºC for several hours [Chem. Commun, 2009, 15, 1970-1971]. Recently, it was observed that a FePc was demetalated during an Oxygen Reaction Reduction process [J. Am. Chem. Soc., 2019, 141, 15684-15692] but in our previous work [ref 29 in the revised manuscript] where FePc-POP was used in the same process, no demetalation was observed which prove the high metal stability of this material. 

Despite these antecedents, we have analyzed the presence of iron in the reaction mixtures by ICP after removing the catalysts and we did not found a relevant amount of metal in the runs, less than 3 ppm.This amount is insufficient to promote the reaction since in the recycling experiments an increase of the ECH conversion was not observed, verifying that there is no leaching of the metal to the reaction and in any case, the unlikely iron derivatives formed are totally inactive in this reaction.

These results and a comment regarding to the stability of the metal in the polymer have been included in the revised version.

Overall, the characterization of FePc-POP/TiOhybrid photocatalyst is poor and it should be improved. UV-vis spectra of FePc-POP, TiOand hybrid FePc-POP/TiOshould be included as a figure in the manuscript. XRD analysis and electron microscopy images with EDX mapping would help to have a better knowledge of the hybrid structure. In fact, authors have included SEM in “Materials and methods” section, but no SEM image is present in the manuscript.

Authors have demonstrated the photocatalytic activity of the FePc-POP/TiO2 hybrid catalyst towards the synthesis of cyclic carbonates using CO2. According to blank experiments, hybrid catalyst features a much higher activity than the individual elements, thus indicating a synergy between them. However, they do not provide a reaction mechanism that clarifies which is the role of each of the components of the photocatalyst. Why is the photocatalytic activity of FePc-POP enhanced when supported on TiO2? What is the role of light in the process? Authors should explain in more detail.

The photocatalytic activity of FePc-POP was included initially in this manuscript only as proof of concept to demonstrate that this polymer could promote this reaction under light. A more in-depth study involving other FePc-POP and TiO2 ratios, solvents, substrates and light powers is in progress and will be published later in a more specialized journal. For this reason we have decided to remove this section from this manuscript.

  • Have the authors measured the temperature of the reactor under illumination? Authors should perform a blank experiment under this temperature and dark conditions to exclude a possible thermal pathway induced by light heating.

Yes the temperature was 30ºC, measured after the reaction was finished. At this temperature under dark no conversion takes place.

  • Is it possible to recycle the FePc-POP/TiO2 hybrid photocatalyst? Authors have not included any recycling test in the study to evaluate the stability of this material. This is from pivotal importance, as stability issues are common in this type of hybrid organic-inorganic structures under reaction conditions.

Yes, we are fully agreed. This is of great importance but as it was previously commented, this work will be expanded and the recycled ones will be published later with the most suitable hybrid and optimized photocatalytic conditions.

The amount of FePc-POP in the hybrid photocatalyst has not been optimized. According to the authors, FePc-POP loading on TiO2 is 10%, have they tried with higher or lower loadings?

Have they quantified the real amount of Fe in the samples

As it was previously commented, this work will be expanded to other FePc-POP and TiO2 ratios to optimize the conditions.

  • Why authors use DMAP as base in the reaction instead of TBAB, as previously reported in the literature? What is the advantage of using DMAP?

In a previous work [ref 12-14 of the revised manuscript], we have demonstrated that DMAP improves the activity of metal-catalyst because generates more electron-rich centers than TBAB which is the main advantage to use it. This behavior was first reported by Paddok et al. [J. Am. Chem. Soc. 2001, 123, 11498–11499] and later works by other authors also confirm that DMAP was a more active base with higher yield compared to other bases, see for example: J. Ing. Eng. Chem., 2015, 24, 98-106; Cat. Sci. Technol. 2016, 6, 3997-4001 or Sustain Engerg. Fuels 2019, 3, 1066-1077.

A comment explaining the selection of this base has been included in the revised manuscript. 

Authors should revise the manuscript and correct some typos and grammatical mistakes:

  • Line 66: “a hybrid catalyst”
  • Line 68: “using a 20 W white LED”
  • Line 69: please define MOF if it is the first time you use this abbreviation.
  • Line 75: “Thus, in this work we report for the first time”
  • Line 124, reaction scheme: The abbreviation of cyclohexene oxide is incomplete. Please exchange “CH” by “CHO”.
  • Line 234: “was stopped”
  • Line 247: Please revise purity of FeCl3, there is a possible typo.
  • Line 253: “were mixed”
  • Line 276: “indicated”
  • Line 277: Please provide lamp specifications.

All mistakes have been corrected 

Reviewer 2 Report

The manuscript by Maya et al. report the CO2 chemical fixation by Iron phthalocyanine based porous organic polymer (POPs). It is interesting to study the potential catalytic properties of those POPs in cycloaddition reaction with CO2, however, the reviewer cannot recommend its publication in Molecules unless the following comments are addressed.

  1. The characterizations of Fe-Pc-POP in table 1 only show numbers without spectra. Some data should be presented as figures in supporting information, such as NMR spectra, TGA, FT-IR, N2 adsorption data and UV-Vis.
  2. In scheme 1, it is not clear how does the iron phthalocyanine connected with phenyl group. Which position is the cross-linking between the two components?
  3. What’s the molecular weight range for this reported POP? Is there any experimental data to show that iron center is Fe2+, not Fe3+ ( such as XPS) ?
  4. Since the control (Entry 5 in table 2) shows that DMAP would facilitate ECH conversion by 30%, I wonder what’s the conversion rate under the ration of 1000:10 (Epoxide: DMAP) without POP.
  5. In order to confirm that FePC-POP acts as a heterogeneous catalyst, It is important to run the ICP-MS of the reaction supernatant to check if there is any Fe leach out to the solution.
  6. Why FePC-POP/TiO2 could act as photocatalyst for ECH conversion? What’s the reaction mechanism?

Author Response

  1. The characterizations of Fe-Pc-POP in table 1 only show numbers without spectra. Some data should be presented as figures in supporting information, such as NMR spectra, TGA, FT-IR, N2 adsorption data and UV-Vis.

Figures of FT-IR, Raman, UV-Vis-NIR, Solid state 13C-NMR, N2 adsorption-desorption isotherms, SEM and EDX images have been included in the supporting information as figures s1-s6. 

  1. In scheme 1, it is not clear how does the iron phthalocyanine connected with phenyl group. Which position is the cross-linking between the two components?

The exact position cannot be determined; this is the reason why the final structure is represented in this way. The most favorable positions in the phthalocyanine rings are the carbons 2,3,9,10,16,17,23 and 24 while in the biphenyl molecules the 4 and 4' positions would be the most susceptible to reaction.

This way of representing these porous polymers is common in polymers obtained with this strategy, see for example: Xu et al. J. Mater. Chem. A, 2015, 3, 1272-1278; Bi et al., Polymer , 2019, 164,  183–190 or Wang et al. Journal of Catalysis, 2017, 355 , 101–109.

  1. What’s the molecular weight range for this reported POP? Is there any experimental data to show that iron center is Fe2+, not Fe3+ (such as XPS)?

They are a class of polymers characterized by their high degree of crosslinking, which generates completely insoluble networks that cannot be characterized by GPC and therefore their molecular weight cannot be determined.

In this work the experimental data that confirms that iron center is Fe2+ is the nuclear magnetic resonance. If the oxidation state was 3+ it would be a paramagnetic compound and the spectrum could not have been recorded.

  1. Since the control (Entry 5 in table 2) shows that DMAP would facilitate ECH conversion by 30%, I wonder what’s the conversion rate under the ration of 1000:10 (Epoxide: DMAP) without POP.

It is likely that the ECH conversion would be higher in the ratio of 1000: 10 (ECH: DMAP). However, this result does not provide the information we need. It should be noted that the reaction has been optimized with lower possible amount of catalyst, being the ratio ECH: DMAP: Fe of 3000: 10: 1 the best conditions found. Therefore, the control experiment in the absence of iron had to be performed under the optimized conditions so as to confirm that they are the best conditions for this system.

  1. In order to confirm that FePC-POP acts as a heterogeneous catalyst, It is important to run the ICP-MS of the reaction supernatant to check if there is any Fe leach out to the solution.

The release of iron from the FePc to the solution would suppose a process of demetalation of this macrocycle that in the conditions of this experiment could not occur. The demetalation reactions of metal phthalocyanines only occurs under certain conditions. For example the demetalation of MgPc can be carried out in a degassed dimethylformamide solution with an excess of 1,4-dihydroxyanthraquinone [Int. J. Phtotoenergy, 2000, 2(1), 9-15]; partial demetalation of NiPc can be achieved by halogenation of using NaBr followed by addition of oxone solution [Fuel, 2015, 161, 43-48] or ZnPc can be also partially demetaled under a mixture of pyridine and HCl at 120ºC for several hours Chem. Commun, 2009, 15, 1970-1971]. Recently, it was observed that a FePc was demetalated during an Oxygen Reaction Reduction process [Chen et al., J. Am. Chem. Soc., 2019, 141, 15684-15692] but in our previous work [ref 29 in the revised manuscript] where FePc-POP was used in the same process, no demetalation was observed which prove the high metal stability of this material. 

Despite these antecedents, we have analyzed the presence of iron in the reaction mixtures by ICP-MS after removing the catalysts and we did not found a relevant amount of metal in the runs, less than 3 ppm. This amount is insufficient to promote the reaction since in the recycling experiments an increase of the ECH conversion was not observed, verifying that there is no leaching of the metal to the reaction and in any case, the unlikely iron derivativees formed are totally inactive in this reaction. These results and a comment regarding to the stability of the metal in the polymer have been included in the revised version. However, hot filtration experiments are the most obvious proof of the heterogeneity of a catalyst.

  1. Why FePC-POP/TiO2 could act as photocatalyst for ECH conversion? What’s the reaction mechanism?

The photocatalytic activity of FePc-POP was initially included in this manuscript only as proof of concept to demonstrate that this polymer could promote this reaction under light. Thus the mechanism that governs this process cannot be proposed at this stage of the investigation. A more in-depth study involving other FePc-POP and TiO2 relations, solvents, substrates and light powers is in progress and when completed, a mechanism may be proposed. For this reason we have decided to remove the photocatalytic section from the revised manuscript and will be published later in a more specialized journal.

Reviewer 3 Report

The authors report heterogeneous catalytic systems based on a polymer of iron phthalocyanin for the formation of cyclic carbonates from epoxides and CO2. They studied the thermal reaction and optimized conditions to get excellent yield and low catalyst loading. They studied the photocatalytic production of the same cyclic carbonate and were able to obtain excellent yield by increasing the catalyst loading and reaction time. They showed that the catalyst has good stability by performing recycling experiments that showed a small decrease in yield over subsequent cycles. The work has been performed in a scientifically sound way and has the potential to be a valuable addition to the field, but some detailed experiments are missing. The manuscript could be accepted several issues have been resolved.

  1. The motivation of the work is unclear from the introduction. The introduction states that many catalysts exist for the reaction, and that even recently developed iron catalysts produce the cyclic carbonate products in high yield and mild conditions. It is therefore unclear what sets the work apart from what is already known. The authors should provide a better motivation for the work in the introduction.
  2. The paper heavily focuses on the conversion of epichlorohydrin. The authors should explain why this compound is important. The other two substrates seem to have limited conversion, which brings into doubt the usefulness of the catalyst.
  3. The authors speak about the turn-over number and frequency of the catalyst, all based on their result with close to quantitative yield of product. The significance of a TON or TOF taken after the reaction has quantitatively converted all the material is very small. For example, the TON depends inversely on the amount of catalyst, as stated by the authors. However, the point of a TON and a TOF is that it should be a property of the catalyst, not the reaction conditions. Therefore, the TON should be determined in conditions where the catalyst is limiting. The same is especially true for TOF. If the TOF is determined after the reaction, it can be easily misleading, as simply doubling the reaction time effectively halves the TOF, even if nothing really happened in the last half of the reaction. The authors should comment on this and especially make it very clear in the table where they compare their catalyst to others, as reporting TON/TOF under very different conditions can be easily misleading. To compare with literature, they should perform an experiment with low reaction time and report the initial TOF at low conversion.
  4. There is little to no explanation about the workings of the photocatalytic reaction. There is lots of precedent for the thermal reaction, but it is unclear to me how light would help in this reaction. The authors should provide an explanation as to why they thought light would perform the reaction.
  5. A control experiment showing that there is no thermal activity in the absence of light is needed. On a related note: did the authors monitor the temperature during the photochemical reaction? The reaction temperature is not stated in Table 4.
  6. The photocatalytic reaction behaves in a strange way. After 24 hours, the conversion is a mere 2%, but an additional 24 hours increases it to 83%. The authors should explain this effect.
  7. The authors claim that the photocatalytic system with TiO2 performs better than the bare FePc-POP system. However, it is not clear that increasing the reaction time further would give higher yield. This experiment is missing and should be performed.
  8. Does the photocatalytic reaction work for the other two substrates?

Minor issues:

The authors should report conversion of substrate as well as yield of cyclic carbonate. If these two are equal in all cases, this should be stated clearly.

The amount of solvent used by the authors in photocatalytic conditions should be described.

Page 4 Table 2 reaction scheme: CH should be CHO, the structure is wrong, and CHO gives a carbonate product that does not correspond to one with one R group.

Page 4 table 2 caption: FePc as catalysts, should be catalyst (singular)

Page 5 line 177: epichlorohydrin is misspelled.

Page 9 line 289: “The ethanol was detached,” it is unclear what the authors mean.

Page 9 line 293: CG-MS should be GC-MS

Author Response

  1. The motivation of the work is unclear from the introduction. The introduction states that many catalysts exist for the reaction, and that even recently developed iron catalysts produce the cyclic carbonate products in high yield and mild conditions. It is therefore unclear what sets the work apart from what is already known. The authors should provide a better motivation for the work in the introduction.

The motivation of this work and a better explanation about what is distinguished from published have been incorporated in the introduction. 

  1. The paper heavily focuses on the conversion of epichlorohydrin. The authors should explain why this compound is important. The other two substrates seem to have limited conversion, which brings into doubt the usefulness of the catalyst.

The reviewer is right; this catalyst is only useful in the conversion of ECH. We have emphasized this in the manuscript and explained the importance of the ECH.

  1. The authors speak about the turn-over number and frequency of the catalyst, all based on their result with close to quantitative yield of product. The significance of a TON or TOF taken after the reaction has quantitatively converted all the material is very small. For example, the TON depends inversely on the amount of catalyst, as stated by the authors. However, the point of a TON and a TOF is that it should be a property of the catalyst, not the reaction conditions. Therefore, the TON should be determined in conditions where the catalyst is limiting. The same is especially true for TOF. If the TOF is determined after the reaction, it can be easily misleading, as simply doubling the reaction time effectively halves the TOF, even if nothing really happened in the last half of the reaction. The authors should comment on this and especially make it very clear in the table where they compare their catalyst to others, as reporting TON/TOF under very different conditions can be easily misleading. To compare with literature, they should perform an experiment with low reaction time and report the initial at low conversion.

We agree with this observation but most of the TOFs included in the preliminary version of manuscript were collected from the corresponding references. We had removed the TOF values and include the epoxide/metal ratios and TON used in each experiment for a better comparison.

  1. There is little to no explanation about the workings of the photocatalytic reaction. There is lots of precedent for the thermal reaction, but it is unclear to me how light would help in this reaction. The authors should provide an explanation as to why they thought light would perform the reaction.

The photocatalytic activity of FePc-POP was initially included in this manuscript only as proof of concept to demonstrate that this polymer could promote this reaction under light. Thus the explanations about this process cannot be proposed at this stage of the investigation. A more in-depth study involving other FePc-POP and TiO2 relations, solvents and light powers is in progress and when completed, a mechanism may be proposed. For this reason we have decided to remove the photocatalytic section from the revised manuscript and will be published later in a more specialized journal.

A control experiment showing that there is no thermal activity in the absence of light is needed. On a related note: did the authors monitor the temperature during the photochemical reaction? The reaction temperature is not stated in Table 4.

The temperature reached in the photocatlytic reaction was 30ºC. In the absence of light there is no reaction.

  1. The photocatalytic reaction behaves in a strange way. After 24 hours, the conversion is a mere 2%, but an additional 24 hours increases it to 83%. The authors should explain this effect.

This behavior can be attributed to the induction period, very long in this system. When we have more experimental data we can offer a concrete explanation for this fact.

  1. The authors claim that the photocatalytic system with TiO2 performs better than the bare FePc-POP system. However, it is not clear that increasing the reaction time further would give higher yield. This experiment is missing and should be performed.

A conversion of 77% in 24h seemed acceptable to us, but we will take this point into account for our next work and we will increase the time.

  1. Does the photocatalytic reaction work for the other two substrates?

As it was previously commented, this work will be expanded to other substrates to see the scope of the reaction

Minor issues:

The authors should report conversion of substrate as well as yield of cyclic carbonate. If these two are equal in all cases, this should be stated clearly.

The cyclic carbonate was obtained as the only product in all reactions, so the yield was equal to the conversion in all cases. It has been specified in the revised manuscript.

The amount of solvent used by the authors in photocatalytic conditions should be described.

Page 4 Table 2 reaction scheme: CH should be CHO, the structure is wrong, and CHO gives a carbonate product that does not correspond to one with one R group.

Page 4 table 2 caption: FePc as catalysts, should be catalyst (singular)

Page 5 line 177: epichlorohydrin is misspelled.

Page 9 line 289: “The ethanol was detached,” it is unclear what the authors mean.

Page 9 line 293: CG-MS should be GC-MS

These mistakes have been fixed

Round 2

Reviewer 1 Report

Authors have come through all the comments, so I think now this work is ready for its publication. 

Congratulations.

Reviewer 2 Report

The authors have addressed my comments. 

Reviewer 3 Report

The authors have corrected the flaws of the paper to my satisfaction.